# CRAAS: A European Cloud Regime dAtAset Based on the CLAAS-2.1 Climate Data Record

**Vasileios Tzallas** [1,*] **, Anja Hünerbein** [1] **, Martin Stengel** [2] **, Jan Fokke Meirink** [3] **, Nikos Benas** [3]**,
Jörg Trentmann** [2] **and Andreas Macke** [1]

[1] Leibniz Institute for Tropospheric Research (TROPOS), Remote Sensing of Atmospheric Processes,
    Permoserstraße 15, 04318 Leipzig, Germany
[2] Deutscher Wetterdienst (DWD), Frankfurter Str. 135, 63067 Offenbach am Main, Germany
[3] Royal Netherlands Meteorological Institute (KNMI), R&D Satellite Observations,
    3731 GA De Bilt, The Netherlands
[*] Correspondence: tzallas@tropos.de

**Abstract:** Given the important role of clouds in our planet's climate system, it is crucial to further improve our understanding of their governing processes as well as the resulting spatio-temporal variability of their properties. This co-variability of different cloud optical properties is adequately represented through the well-established concept of cloud regimes. The focus of the present study lies on the creation of a cloud regime dataset over Europe, named "Cloud Regime dAtAset based on the CLAAS-2.1 climate data record" (CRAAS), in order to analyze their variability and their changes at different spatio-temporal scales. In addition, co-occurrences between the cloud regimes and large-scale weather patterns are investigated. The CLoud property dAtAset using Spinning Enhanced Visible and Infrared (SEVIRI) edition 2.1 (CLAAS-2.1) data record, which is produced by the Satellite Application Facility on Climate Monitoring (CM SAF), was used as the basis for the derivation of the cloud regimes over Europe for a 14-year period (2004–2017). In particular, the cloud optical thickness (COT) and cloud top pressure (CTP) products of CLAAS-2.1 were used in order to compute 2D histograms. Then, the k-means clustering algorithm was applied to the generated 2D histograms in order to derive the cloud regimes. Eight cloud regimes were identified, which, along with the geographical distribution of their frequency of occurrence, assisted in providing a detailed description of the climate of the cloud properties over Europe. The annual and diurnal variabilities of the eight cloud regimes were studied, and trends in their frequency of occurrence were also examined. Larger changes in the frequency of occurrence of the produced cloud regimes were found for a regime associated to alto- and nimbo-type clouds and for a regime connected to shallow cumulus clouds and fog (−0.65% and +0.70% for the time period of the study, respectively).

**Keywords:** CRAAS; cloud regimes; Europe; climate; variability; weather types

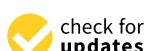



## 1. Introduction

Cloud variability has been a very important field of research due to the fact that clouds play a major role in the Earth's water cycle and energy budget. Changes in cloud optical properties and cloud cover could result in significant changes in the Earth's climate system, through feedback mechanisms, by affecting the hydrological cycle and the radiative balance. Therefore, constant monitoring of clouds and their variability, on both a global and regional scale, is essential. This has been assisted by remote sensing techniques and the ability to create long-term and uniform satellite cloud data records.

In order to properly monitor and understand the complexity of clouds and their resulting feedback mechanisms, various classification methods have been developed since several studies [1–7] have supported the idea that traditional cloud types cannot properly represent an atmospheric scene. For instance, over a certain region for a given time period there might be a governing cloud type, but other co-occurring cloud types are encountered

as well in that region for that time period [6]. This introduces the need for "weather state" or "cloud regime" studies as they can represent the predominance of mixtures of different cloud types over a region for a certain time period.

Cloud regime analysis has been considered useful and adequate for various types of studies. To list a few, according to Rossow et al. in [2], cloud regimes can be useful in order to study convective systems and to compute transition probabilities between the distinct regimes, as well as for the better representation and diagnostics of many global weather and climate models. In addition, one of the main types of studies in which cloud regimes have been used so far is the calculation of regime radiative effects [5,6]. Finally, cloud regimes can be applied for climate monitoring both on a global scale and for the regional long-term variability of climate. To assist with the last type of study, cloud regimes have been found to be more robust to small, statistically insignificant, cloud property variations since the morphology of the full range of cloud properties is accounted for. These small variations could otherwise lead to misinterpretations of the long-term variability [6,8,9].

For the derivation of cloud regimes, both active [10] and passive [1–7] satellite remote sensing observations have been used in the past, each of them providing a different insight into the study of clouds, based on the advantages and disadvantages that arise from each observation type. Active instruments provide better representation of the vertical distribution of different overlapping cloud types, while passive instruments have the ability to capture the simultaneous existence of different cloud types in a large area. In addition, polar orbiting and geostationary satellites can assist in distinct ways in cloud regime studies. Polar orbiting imagers allow us to have cloud property information on a global scale, but the temporal resolution is more limited compared to that of imagers on a geostationary orbit. Therefore, the use of observations from instruments on a geostationary orbit allow for a more detailed sampling of the diurnal cycle of the cloud properties.

It has been shown that different cloud types can have different variabilities throughout the day because of the distinct processes of their formation, evolution and dissipation. It has also been observed that various cloud types behave in distinct or even opposite ways, over land and sea [11]. For example, Eastman and Warren in [12] studied how cumulus, cumulonimbus, stratus, stratocumulus and fog vary over land and ocean during the day using surface observations. They found that each cloud type behaves in a certain way, no matter the climate regime. The diurnal cycle over land areas is mostly dominated by cumuliform clouds during summer and stratiform clouds in winter. Sea areas exhibit a smaller amplitude of the diurnal cycle, in comparison to continental areas, and are primarily driven by stratiform clouds. These changes in cloud types can be captured by instruments on geostationary satellites, e.g., the SEVIRI instrument on Meteosat Second Generation (MSG) satellites [13], which have a repeat cycle of 15 min. Consequently, data from geostationary satellites can prove to be very useful in order to include the diurnal variability of cloud properties into cloud regime analyses and, also, to study how the distinct cloud regimes evolve and behave during the day.

Previous studies of cloud regimes have focused mainly on the tropics [1–3,8,14–17], on the Southern Ocean [18,19] or on a global scale [4–7,20–22]. The earliest of these studies mainly used cloud property datasets from the International Satellite Cloud Climatology Project (ISCCP) [1–4,8,14,18–20]. In more recent years, different collections of the Moderate Resolution Imaging Spectroradiometer (MODIS) have been utilized [5–7,15–17,21,22]. In the present work, cloud regimes are derived from the CLAAS-2.1 (CLoud property dAtAset using SEVIRI (Meteosat Spinning Enhanced Visible and Infrared Imager), edition 2.1) data record over Europe (Figure 1) from 2004 to 2017. This provides a unique opportunity to use a long-term, updated and validated satellite data record to study the variability of cloud regimes over a region (Europe and the Mediterranean), which has been characterized as a climate hot spot [23,24], and therefore a constant climate monitoring of the variability of the cloud properties of this region is of high importance. In addition to that, Leinonen et al. in [7] found that there is a region-dependent variability within each regime, resulting from the different meteorological environments, thus making it more useful and necessary to

study the variability of cloud regimes over such a region. Smaller or more apparent differences of the produced cloud regimes in this study, compared to the ones of other studies, are to be expected given the different areas of interest and also because of differences in the underlying observations, the algorithms of cloud detection and cloud optical property retrievals, and the sampling and gridding approaches [22].

The long-term variabilities of the cloud regimes, along with their annual and diurnal cycles, are examined. More regional aspects of the regime variability are studied by selecting a smaller region, covering mostly Germany, within the European domain. The European region is defined from 30°N to 60°N and from 11°W to 37°E, while the region of Germany extends from 45°N to 55°N and from 3°E to 16°E. Over this sub-region, the notion that certain cloud types are associated to specific large scale weather patterns is investigated through a comparison between the derived cloud regimes and the Objective Weather Type Classification (OWTC) [25], which is produced by the German Meteorological Service (DWD).

A description of the CLAAS-2.1 data record, which is used in this study, along with the applied methodology and the generated Cloud Regime dAtAset based on the CLAAS-2.1 (CRAAS) dataset, is provided in Section 2. The derived cloud regimes, together with their spatio-temporal variability and the additional analysis are included and discussed in Section 3. The present study is then summarized in Section 4, where an overview of the main conclusions is also provided.

## 2. The CRAAS Dataset and Methodology

### 2.1. Description of the CLAAS-2.1 Climate Data Record

The CLAAS-2.1 data record is a recent long-term cloud property data record, which is produced by the EUMETSAT CM SAF (Satellite Application Facility on Climate Monitoring) and is an improved and extended version of CLAAS-1 [26,27]. Cloud property algorithms are applied to retrieve geophysical parameters from inter-calibrated measurements of SEVIRI, which is mounted on four Meteosat Second Generation (MSG) satellites, of which MSG-1, MSG-2 and MSG-3 have been used in CLAAS-2.1. The MSGv2012 software package by the NWCSAF (Satellite Application Facility for support to Nowcasting and Very Short Range Forecasting) is applied for the derivation of cloud mask (and cloud fraction) and cloud top properties [28]. In particular, for the derivation of cloud top pressure (CTP), different approaches are used, including a best fit between the simulated and the measured 10.8 μm brightness temperatures, the infrared window intercept method [29] and the radiance rationing method [30]. Additional information on the implementation of the retrieval algorithm for the cloud properties is available in [26] and in [31]. For the retrieval of cloud optical and microphysical properties, the CPP (Cloud Physical Properties) algorithm [32,33] is additionally used. The CPP algorithm retrieves the cloud optical thickness (COT) and the cloud effective radius from the non-absorbing 0.6 μm and the absorbing 1.6 μm channels of SEVIRI by applying the Nakajima and King [34] approach, and is based on lookup tables of top-of-atmosphere reflectances simulated by the Doubling-Adding KNMI (DAK) radiative transfer model [35]. The aforementioned products of the climate data record are available in different spatial and temporal resolutions through the CMS AF portal. A more detailed description of the CLAAS-2.1 climate data record can be found in [27] and the corresponding Algorithm Theoretical Basis Documents (ATBDs) in [31,33].

### 2.2. Derivation of CRAAS

In this study, the level-2 pixel-level products from the CLAAS-2.1 climate data record were used from 2004 to 2017 for the derivation of the joint cloud histograms (JCH) of CTP and COT, which form the basis for the derivation of the cloud regimes. The JCHs consist of six and seven bins of COT and CTP, respectively, which can form a 42-dimensional vector. The specific bin edges of the 2D histograms are 0.0, 1.3, 3.6, 9.4, 23.0, 60.0, 150.0 for COT and 0, 180, 310, 440, 560, 680, 800, 1000 for CTP. The same COT and CTP bin widths as in relevant ISCCP and MODIS cloud regime studies were used in order to have visually

comparable results. Additionally, the cloud mask (CMA) product from CLAAS-2.1 was used for the calculation of the cloud fraction. First, instantaneous (15-min cycle) data of CTP and COT were grouped in $1° \times 1°$ spatial resolution grid cells. The nadir spatial resolution of SEVIRI is $3 \times 3$ km$^2$ leading to approximately 1369 raw pixels in one grid cell. The spatial resolution is further decreased with an increasing distance from the nadir satellite point. The precise numbers of cloudy, clear and valid pixels in each $1° \times 1°$ grid cell were stored in the CRAAS dataset. Then, quality flags from the CLAAS-2.1 climate data record were used to determine the validity of the information in each grid cell and to assist in the selection of the JCH during the derivation process of the cloud regimes. If more than 90% of the pixels were flagged good or valid (based on the CLAAS-2.1 quality flags) for the three variables COT, CTP and CMA, then the corresponding $1° \times 1°$ grid cell was flagged accordingly. In addition, for the valid CMA grid cells, a cloud fraction variable was generated by dividing the number of the pixels classified as cloudy (as defined by the CMA product from the CLAAS-2.1 data record) by the number of the available valid pixels (i.e., all the pixels where the CMA information is available) within the same $1° \times 1°$ grid cell. Finally, the joint cloud histograms were generated by storing the counts of valid pixels, normalized to the sum of valid and clear pixels, into the joint cloud histogram for each grid cell.

In total, during the covered time period and over the region of this study, the number of available data points exceeds 700 million. Approximately 44% of these data points have valid cloud optical thickness information, with the corresponding number for the cloud top pressure being close to 99%. This large difference results from the requirement of daylight conditions for the cloud optical thickness retrieval. However, with cloud optical thickness being essential for the derivation of the joint cloud histograms, the night time slots could not be taken into account.

After generating the joint cloud histograms for the full dataset, the following process was followed in order to derive the cloud regimes. All the joint cloud histograms for which the corresponding daylight COT, CTP and CMA products were available and valid were selected using the respective quality flags. These selected joint cloud histograms account for 310,382,812 out of a total number of 706,959,360 available data points. This availability of the joint cloud histograms is widely regulated by the existence and validity of the cloud optical thickness as well as by the required illumination conditions. Given the large amount of data, sampling the selected JCHs for practical reasons was required. To support that, a number of sensitivity studies on the sampling method, which took into account randomly selecting different sizes of samples of the valid data points, using daily means, or hourly aggregates, indicated that the produced cloud regimes, as evaluated by the pattern correlation coefficients of their centroids and their mean cloud optical thickness, cloud top pressure, cloud fraction and relative frequency of occurrence properties, did not show any significant change when the sampling size was changed. Similar findings on the robustness of the method are mentioned in similar studies, as, for example, in [19]. To avoid biases induced by the dissimilar data availability, which is produced by different geographical patterns of the illumination conditions (e.g., more data availability over the lower latitudes), and in order to facilitate the utilization of the substantial data volume, a sampling scheme was applied to the joint cloud histograms. To complement that, weights accounting for the changing area moving poleward and for the different length of the month were applied as well. The acquisition of an equal-sized sample for each year, to avoid biases due to changes in the frequency of failed retrievals, was also taken into consideration.

The process of generating and selecting the joint cloud histograms from the CLAAS-2.1 climate data record provides the data points that serve as the basis for deriving the cloud regimes and, by extension, the CRAAS dataset. The cloud regimes are produced by applying the k-means clustering algorithm [36] to the selected joint cloud histograms of cloud optical thickness and cloud top pressure. The k-means algorithm, in essence, consists of four iterative steps: (1) k (which is a predefined number) randomly selected data points are used as the initial cluster centers, also called centroids; (2) each one of the data points is

assigned to the nearest cluster center (centroid) by calculating the Euclidean distance to each centroid; (3) new centroids are found by taking the average of the assigned data points of the previous step; (4) the second and third steps are repeated until none of the cluster assignments change. More specifically, for the case of cloud regime studies, each individual joint cloud histogram for every grid cell and every time step is a potential data point, which form a 42-dimensional feature vector [1], for the k-means clustering algorithm. The outcome of the algorithm, after convergence is achieved, is the development of centroids in the form of joint cloud histograms of seven cloud top pressure and six cloud optical thickness bins. In principle, a centroid can be considered as the representative or mean joint cloud histogram of its cluster and, by extension, the representative histogram of the cloud regime.

To determine the optimal number of clusters, several metrics exist [22]. However, most of these metrics are useful for data that are relatively easily clustered, and some of these result in a number of clusters larger than what is practically needed in cloud regime studies [22] and for what is needed to describe the nature of the clouds behind these cloud regimes. As a result, for the purpose of defining the desired number of cloud regimes, a similar concept to that of previous cloud regime studies, e.g., [1–5], was followed. A slightly different process for determining the final number of cloud regimes was used only in the more recent versions of the MODIS-based cloud regimes, as for example in [6], where they used nested clustering starting with a relatively small number of core cloud regimes and then splitting them into several sub-regimes. Several trials starting with a small number of clusters (k) and then increasing that number while examining whether criteria on, mainly, the pattern correlation coefficients between different centroids were met and how the mean properties of the produced cloud regimes were changing. In particular, the resulting centroid histogram patterns should not change significantly (as judged by the pattern correlations among the centroids) and the resulting centroid patterns should differ from each other significantly (pattern correlation coefficients should be low, usually less than 0.8). The pattern correlation coefficients were selected subjectively, as there is no specific criterion to define them, and as it is usually performed in cloud regime studies, e.g., [6]. Additionally, subjective criteria on the decision of the number of clusters, regarding the nature of the clouds represented through the cloud regimes, were applied. To be more precise, during the derivation process, while increasing the number of the desired cloud regimes (CR), the pattern correlation coefficients remained well below 0.8 until the case of k = 8; for the case of k = 8, there was an increased correlation (0.87) between CR6 and CR8. However, given that for smaller values of k, CR5 and CR8 were mixed together, and that k = 8 was the first case where a separation of the two CRs was observed, subjective criteria were taken into account in the decision to select the case of k = 8 because the characteristics (in terms of the nature of the clouds represented by these CRs) of the two CRs (CR5 and CR8) were important enough to be maintained in discrete CRs.

After the centroids of the cloud regimes were calculated, all the JCH histograms generated from CLAAS-2.1 for every 15-min time step and every 1° × 1° grid cell were assigned to one of the cloud regimes. To achieve that, the Euclidean distance between every COT–CTP histogram and the centroid of each cloud regime was calculated. Then, the cloud regime with the minimum Euclidean distance was assigned to each JCH.

The CRAAS dataset contains the generated 2D COT–CTP histograms along with the centroids of the derived cloud regimes and the data points labeled by the cloud regime to which they are classified, among additional data such as the COT and CTP bins, and the applied quality flags. Data from CRAAS are available in netCDF4 format, following the Climate and Forecasts 1.8 (CF-1.8) Conventions and can be accessed through [37].

## 3. Results and Discussion

### 3.1. The CRAAS Cloud Regimes

3.1.1. Overview of the Cloud Regimes

Following the methodology described in Section 2, a set of eight cloud regimes was derived from the CLAAS-2.1 data over Europe and for the time period of 2004–2017. The corresponding centroids of the eight derived cloud regimes, in the form of joint cloud histograms of cloud top pressure and cloud optical thickness, are shown in Figure 1. In order to facilitate a quick visual comparison to previous cloud regime studies, the cloud regimes are presented in an order, starting from the ones containing the largest fraction of high-level clouds and continuing with those that consist of lower-level clouds. For each joint cloud histogram, the bins of cloud top pressure are located in the y-axis and those of cloud optical thickness are on the x-axis. The cloud fraction of the respective bins is represented by the color bar. The total cloud fraction of each cloud regime, as well as the regime's overall mean relative frequency of occurrence (RFO) are presented on top of the corresponding plot. Finally, the geographical distribution of the multiannual mean RFO of the eight CRs is presented in Figure 2. An overview of the CRs, including their mean properties and a name based on the cloud types mainly associated to each CR, can be found in Table 1.

In comparison to other cloud regime studies, various similarities in the patterns of the centroids or in the maps of the multiannual mean RFO can be observed. However, differences can be noticed as well. First of all, the number of the derived cloud regimes could change depending on several steps of the method and, more importantly, because of the selected region of each study. For instance, for cloud regime studies that focus on smaller regions (compared to global studies, for example), the number of regimes required to represent the cloud conditions over that area is naturally expected to be smaller. Compared to global studies, it can be noted that the cloud regimes of the present work have a smaller representation of the very high clouds that would fall in the first bin of cloud top pressure. This can be justified by considering that in the global studies, the tropics are included and, as a result, regions with much more convective systems and with higher tropopause levels, and by extension with higher clouds, are taken into account. For instance, the CTP bins with values less than 180 hPa contain considerably fewer clouds compared to the corresponding bins of the global studies (e.g., [4,6]) that include the Intertropical Convergence Zone (ITCZ). Lastly, differences could be attributed to discrepancies in the observations, the algorithms of cloud detection and cloud optical property retrieval, and the sampling and gridding approaches [22]. Consequently, comparing cloud regimes can be a very challenging task that requires consideration of several parameters.

The patterns appearing in the centroids confirm that the cloud regimes contain mixtures of different cloud types. However, distinct peaks of the cloud fraction in the centroids can lead to the association of the clusters to certain, more dominant, cloud types in this apparent composition of cloud types [22]. Following this notion, a description of the derived cloud regimes and an attempt to associate the cloud regimes to certain known cloud types, also taking into consideration their mean geographical distribution, is presented below.

Cloud Regime 1 (CR1) contains a large amount of high and thin clouds; it has a total cloud fraction of 82.9% and a small overall mean RFO of 4.9%. Generally, it has a low RFO over most of Europe; however, there are a few specific areas, mostly near the coasts, of slightly higher RFO (e.g., over the Gulf of Lion and the Ligurian Sea, the Adriatic Sea and the Baltic Sea). As a result, CR1 can be connected mostly to cirrus and, because of the relatively reduced cloud fraction compared to the other high-level CRs, to cirrocumulus clouds. CR2 presents its peak of cloud fraction in the bins of low cloud top pressure with the cloud optical thickness ranging from low to medium values. A few deeper clouds are present in this cloud regime as well. Moreover, it has a larger total cloud fraction (97%) compared to CR1, and its overall RFO is again small (4.4%). The geographical multiannual mean RFO is also low over the whole region, but it shows slightly more continental characteristics than CR1. Consequently, CR2 is associated with cirrus and

cirrostratus clouds. The last regime of those with the higher clouds is CR3. This CR has some of the thickest clouds out of all the eight derived regimes and the highest total cloud fraction (99.7%). However, its overall mean RFO (4%) is the lowest of every other CR. The map of its multiannual mean RFO has very low values over the full region, especially over the lower latitudes. Only an area over Central and Eastern continental Europe shows a faint increase in the RFO values. Cirrostratus and clouds related to deep convection can be attributed to CR3; CR3 is the cloud regime that mostly represents convection and storm systems over Europe.

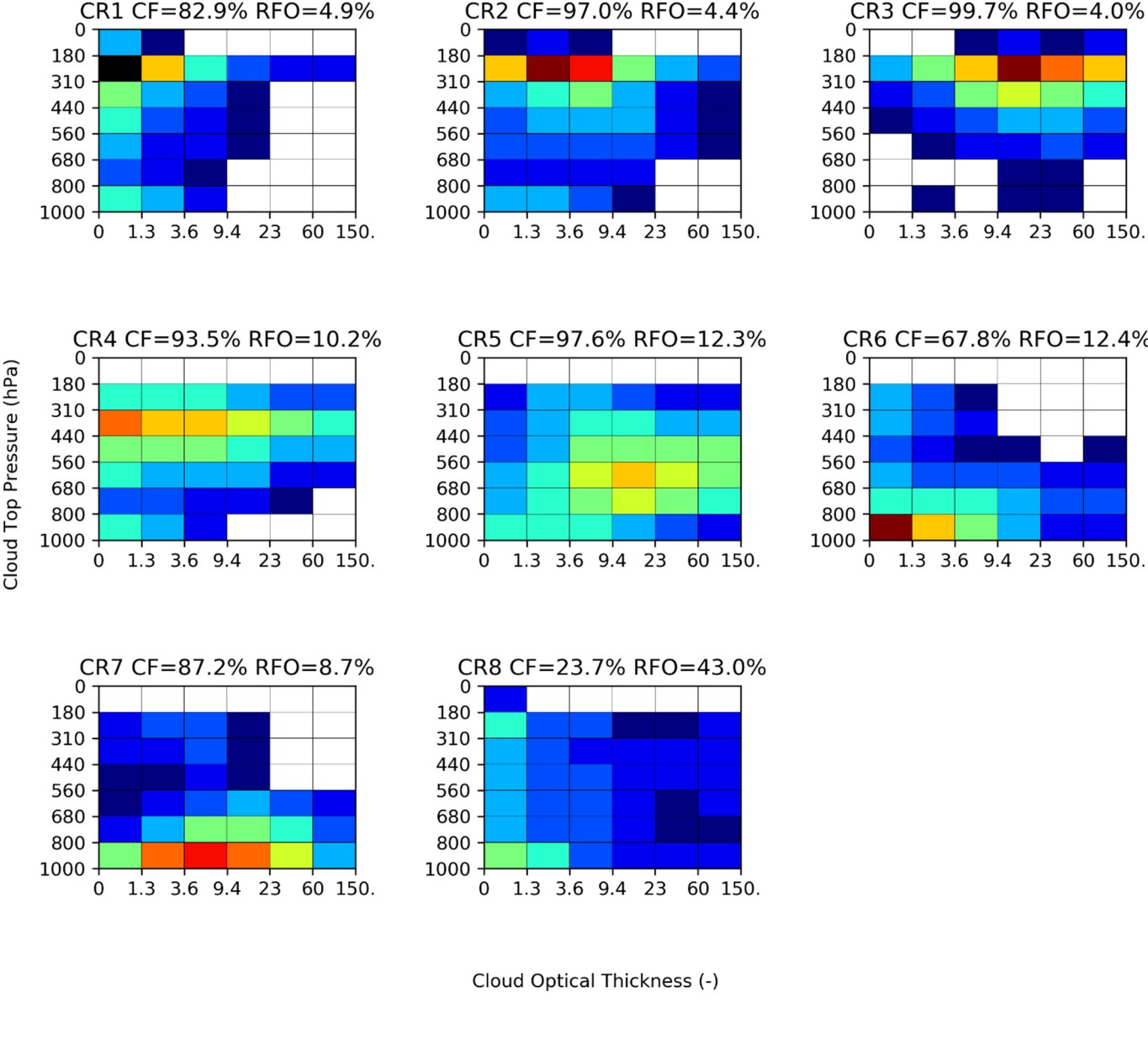

**Figure 1.** Centroids of the 8 CRAAS cloud regimes (CRs). The CRs were derived over Europe during the 2004–2017 period. On top of each centroid the total cloud fraction (CF) and the overall mean relative frequency of occurrence (RFO) is shown.

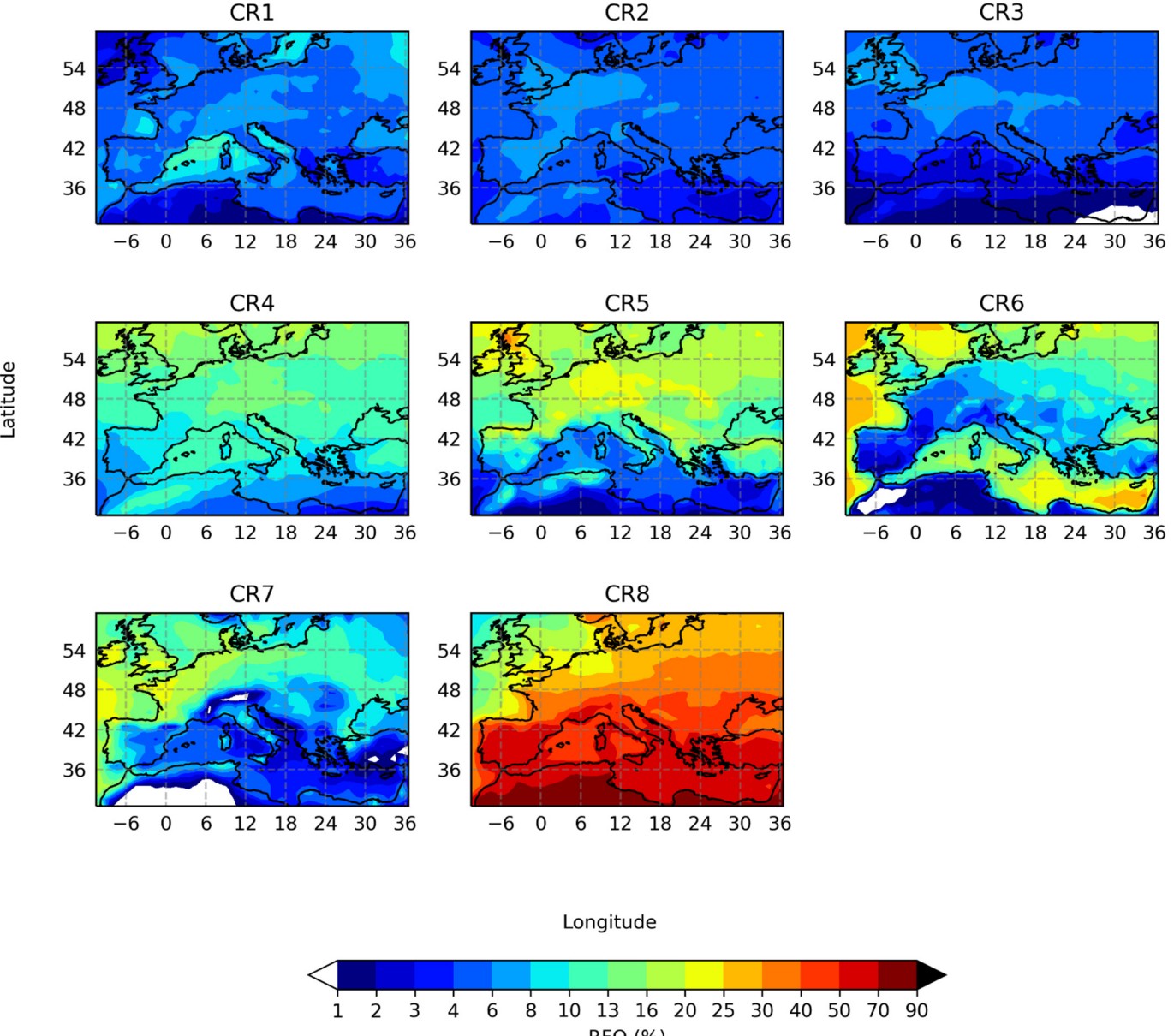

**Figure 2.** Geographical distribution of the overall mean relative frequency of occurrence (RFO) of the 8 CRAAS cloud regimes (CRs).

An additional CR connected to cirrus, similar to CR1 but with its peak of cloud fraction taking place in bins of larger cloud top pressure, and also with a significantly wider range of cloud top pressure and cloud optical thickness, is CR4. The pattern of CR4 also indicates the existence of altocumulus and altostratus clouds in CR4. The total cloud fraction of CR4 is relatively high (93.5%), and it is a much more frequent CR than the three high-level CRs, with an overall RFO value of 10.2%. CR4 presents an increased RFO over, generally, most of the region, with decreased RFO values at lower latitudes. Cloud fraction peaks in bins of higher cloud top pressure for CR5, in comparison to the previous CRs. It contains a variety of middle-level clouds, altostratus and nimbostratus clouds. In general, clouds associated to CR5 present a widely ranging cloud optical thickness, extending mostly from medium to high values. It is one of the most cloudy regimes, in terms of total cloud fraction (97.6%), and has a higher overall mean RFO (12.3%) than the previous CRs. It is a cloud regime with a clear continental character, showing larger RFO values over land areas, and it also presents generally increased RFO values over higher latitudes. Over lower latitudes

it appears to be less frequent, with the lowest values taking place over the Mediterranean Sea. The areas that CR5 is more frequent in, appear to be Germany and the Alps, and parts of the northern United Kingdom.

**Table 1.** Summary of the 8 CRAAS cloud regimes (CRs) including the mean properties and the main cloud types associated to each of the cloud regimes.

| Cloud Regime | Mean COT (-) | Mean CTP (hPa) | CF (%) | RFO (%) | Main Cloud Types |
|---|---|---|---|---|---|
| CR1 | 2.4 | 308.1 | 82.9 | 4.9 | Cirrus |
| CR2 | 7.0 | 302.3 | 97.0 | 4.4 | Cirrostratus |
| CR3 | 31.7 | 286.3 | 99.7 | 4.0 | Dense cirrus from deep convection and from frontal systems |
| CR4 | 12.0 | 425.3 | 93.5 | 10.2 | Alto- and nimbo-type clouds |
| CR5 | 29.5 | 614.8 | 97.6 | 12.3 | Mid-level clouds |
| CR6 | 5.2 | 861.3 | 67.8 | 12.4 | Shallow cumulus, fog |
| CR7 | 14.2 | 882.7 | 87.2 | 8.7 | Stratocumulus |
| CR8 | 12.2 | 627.6 | 23.7 | 43.0 | Fair-weather clouds |

The following two CRs, namely CR6 and CR7, are cloud regimes consisting mainly of the lowest clouds of all the CRs. They differentiate from each other mostly on the cloud optical thickness bins where the respective total cloud fraction presents its largest values. CR6 contains mostly low and very thin clouds. It has a significantly reduced total cloud fraction value (67.8%) and an overall mean RFO of 12.4%. CR6 has the most pronounced oceanic character out of all the cloud regimes, presenting its highest values of RFO over the Atlantic Ocean, the Bay of Biscay, the North Sea and the Mediterranean over the Libyan and the Levantine Sea. The low cloud optical thickness, along with the reduced total cloud fraction and the oceanic character of CR6, indicate that this CR consists primarily of shallow cumulus clouds and, potentially, fog. CR7 contains optically thicker clouds than CR6, which also have a much higher total cloud fraction (87.2%) and a slightly reduced RFO (8.7%). The areas of maximum RFO are found to be in the Atlantic Ocean and the North Sea as well as coastal and some more inland areas of Central and Western continental Europe. Over lower latitudes the RFO of CR7 is significantly reduced. Therefore, CR7 can be associated to marine stratocumulus and to some broken stratocumulus clouds.

The last cloud regime, CR8, is the hardest cloud regime to associate with any specific cloud type. It consists of all the joint cloud histograms that do not have any distinguishable pattern in order to suggest a particular identity [6]. CR8 has the highest overall mean RFO (43%) and the lowest total cloud fraction (23.7%). It is, therefore, a cloud regime that appears very often but it contains clouds with low cloud fractions. It has a very high RFO over the whole region, and its RFO decreases slightly only over some very cloudy areas (mostly marine), where more frequent, cloudier CRs (e.g., CR4 to CR7) take over. As a result, CR8 can be described as a fair-weather cloud regime [38]. Interestingly, a similar cloud regime has been found in every previous cloud regime study, e.g., [4,6,22,38], regardless of the region of interest.

The cloud regime classification is illustrated in Figure 3 using an RGB image (Figure 3a) and a color-coded map of the cloud regimes (Figure 3b) for 1 February 2014 at 10:30 UTC. This time step serves as a good example for the cloud regime classification as a wide variety of cloud types is apparent over the region of interest. The majority of the different features of this dynamic field are well represented by the CRAAS dataset by attributing each $1° \times 1°$ grid cell to the corresponding CR. However, given the coarser resolution of CRAAS, in general, cases of misclassification due to small scale variability of cloud types can also be present. The broken low-level clouds over the Atlantic Ocean are well classified to CR6, and slightly more to the east, where the clouds become higher, the shift to the alto-

and nimbo-type clouds of CR4 can be noticed. An additional well-represented feature is the transition of the cloud regimes for the frontal system over France; observing from west to east, the change from high thin cirrus (CR1) to cirrostratus (CR2) and then to more convective high and optically thick clouds (CR3) can be seen. Finally, in general, the areas over which only some scattered clouds exist and where the cloud fraction is significantly reduced are attributed to the fair-weather cloud regime (CR8).

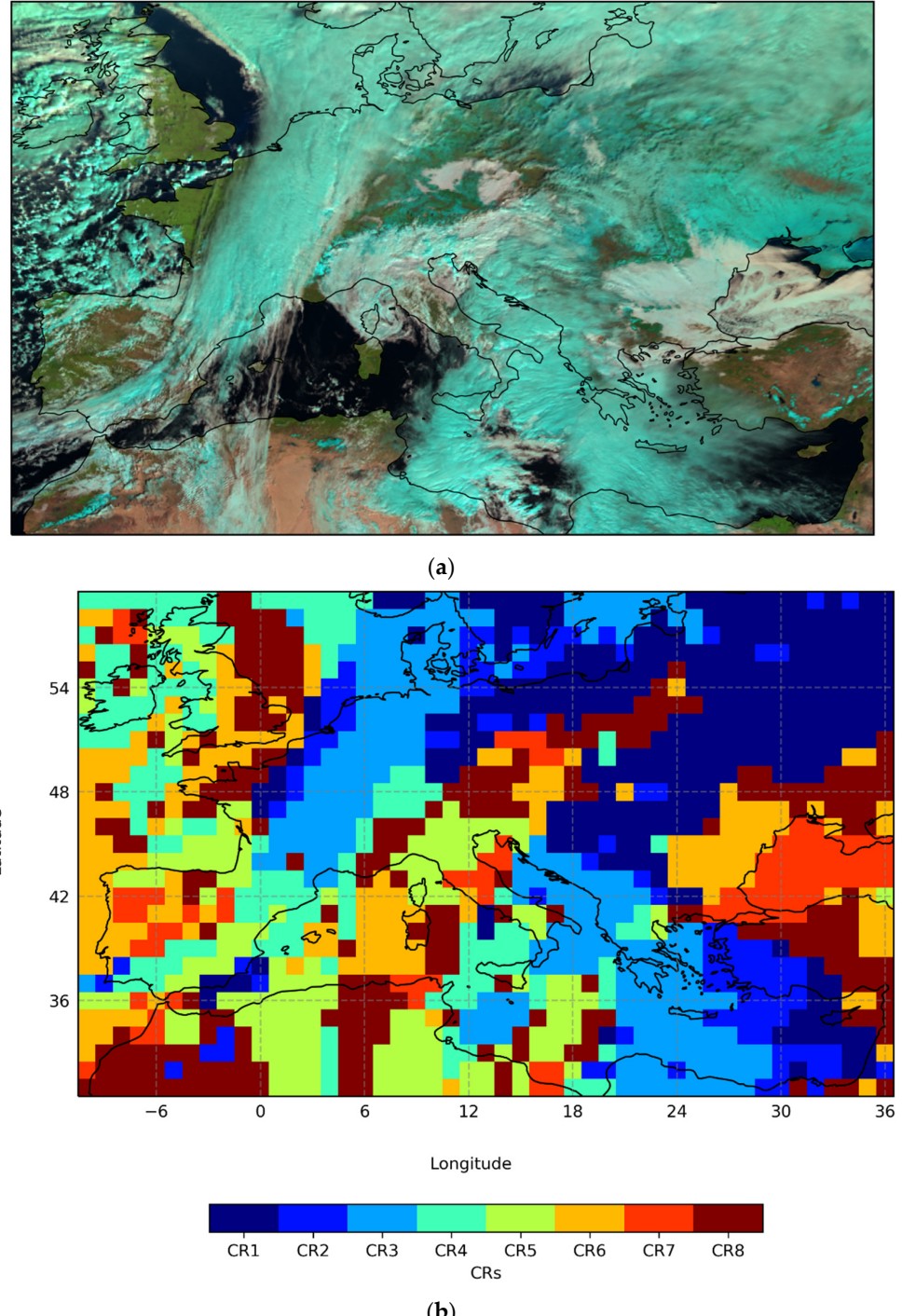

**Figure 3.** Illustration of the cloud regime classification from CRAAS for 1 February 2014 at 10.30 UTC. (**a**) SEVIRI Natural Color RGB image over Europe, composed using the 0.6, 0.8 and 1.6 micron channels; (**b**) color-coded map of the CRAAS cloud regime classification.

3.1.2. Variability of the Cloud Regimes over Europe

In order to have a more in-depth understanding of the nature and the behavior of these eight cloud regimes, a study of their variability at different time scales was carried out. Given that the CRAAS dataset covers an extensive period (from 2004 to 2017), the variability of the cloud regimes provided below can be thought of as a climatology of these cloud regimes over Europe. At the same time, an opportunity to observe long-term changes in each CR and, by extension, in the clouds over Europe is provided here.

The overall annual cycle of the RFO of each CR is displayed in Figure 4. The intra-annual variability is clearly dominated by the fair-weather CR8. The monthly mean RFO of CR8 increases as one moves from winter to spring, peaks in the summer months reaching its maximum of 54.2% in July, and then decreases during autumn. The wide annual cycle of CR8 is also clear in the seasonal mean RFO; for summer the overall mean RFO is approximately 51.1%, which then decreases considerably to 29.3% for the winter season. This annual cycle is mostly influenced by the southern regions of Europe where these low cloud fraction scenes are much more frequent, particularly in summer. The three CRs associated to high-level clouds (i.e., CR1, CR2 and CR3), which are the ones that have the lowest overall mean RFO, present a quite uniform annual cycle of their RFO, with a very small variability. CR1 and CR2 in particular have a nearly identical annual cycle, presenting two small but distinguishable peaks in May and September–October. CR3, being widely connected to clouds from frontal systems, has the lowest frequency of occurrence of all the CRs throughout the year, but it starts becoming more frequent with the start of autumn leading up to winter, which is when frontal systems become more frequent over Europe. As expected, CR3 shows its minimum value in June (3.5%), and it reaches its maximum value in December (6.3%). The next three regimes (i.e., CR4, CR5 and CR6) have more pronounced annual variabilities of RFO than the three high-level cloud regimes. The alto- and nimbo-type cloud regime, CR4, shows its highest mean RFO values in winter (seasonal mean RFO 15.7%) and this decreases to approximately half of that value during summer (seasonal mean RFO 7.8%), where it presents its minimum. The middle-level regime, CR5, takes its maximum RFO values in spring. In fact, for that season, CR5 is the second most dominant cloud regime, after CR8. The shallow cumulus, broken marine stratocumulus clouds and the fog included in CR6 are considerably more frequent during autumn and especially during winter (seasonal mean RFO 18%), where CR6 is the second most frequent CR, but present significantly lower RFO values during summer (seasonal mean RFO 7.7%). Finally, the annual RFO variability of CR7 has much lower RFO values than the other low-level CR, CR6, during autumn and winter, and it presents only some moderate fluctuations during the course of the year.

A number of studies on the diurnal cycle of clouds using different satellite datasets exist, e.g., [39–43]. Long-term surface observations have also contributed significantly to the topic; for instance, Eastman and Warren in [12] extensively describe the diurnal cycle of different cloud types on a global scale using ground-based observations. Here, taking advantage of the high temporal resolution of the SEVIRI measurements, which allows the CLAAS-2.1 climate data record to provide cloud optical properties every 15 min, the CRAAS dataset offers a unique opportunity to investigate the diurnal variability of clouds through the concept of cloud regimes for the first time.

The diurnal variability of each of the eight cloud regimes derived from CRAAS, averaged over the whole region of study for hourly true solar time bins, is shown in Figure 5. In a similar manner to the annual variability, the fair-weather cloud regime CR8 is the most dominant in terms of RFO values. It presents a peak in the early morning, reaching RFO values of 51.5%, and then decreases during the day, while the majority of the rest of the cloud regimes gradually become more frequent, and, finally, a second peak of approximately the same magnitude as the first one is apparent in the evening.

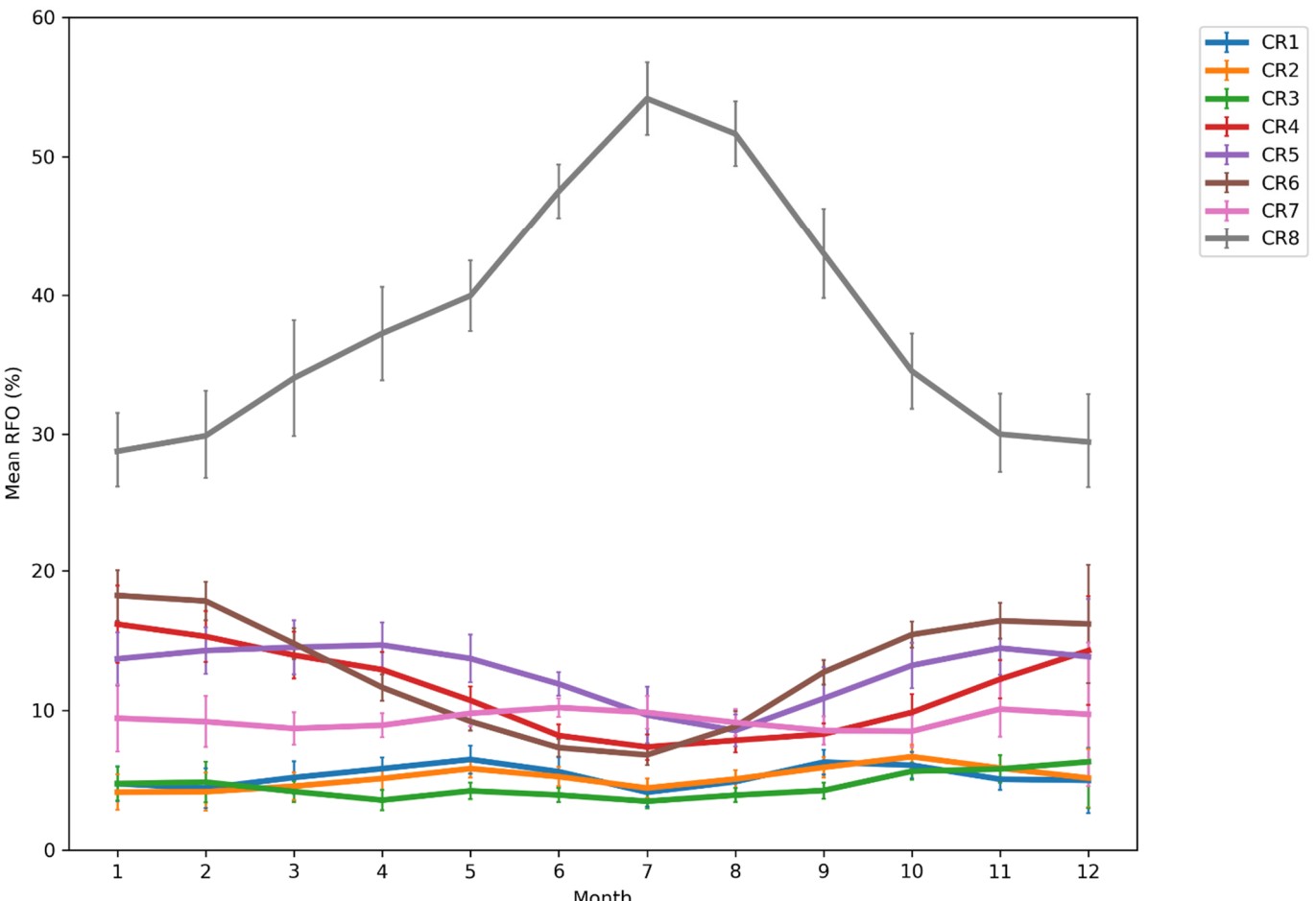

**Figure 4.** Mean intra-annual variability of the relative frequency of occurrence (RFO) of the 8 CRAAS cloud regimes (CRs) over Europe for the 2004–2017 period.

Regarding cloud regimes containing mainly low-level clouds (mostly CR6 and CR7), as expected, their diurnal cycle appears to be driven primarily by destabilization of the boundary layer by solar heating of the surface, inducing convection and leading to the formation of cumuliform clouds. After the boundary layer starts cooling, moving more towards night times, condensation is caused and more stratiform clouds are generated. Following this principle, the peak of the shallow cumulus clouds of CR6, with values up to 14.4%, can be observed slightly after noon. The increased RFO values in the early morning and late evening can be attributed to fog, also being represented in this cloud regime, which is more frequent near sunrise; then it sharply decreases until early evening. Typically, stratocumulus and stratus clouds, represented mainly by CR7, are more frequent in the early morning (showing a maximum 10.8%), drastically decreasing after noon.

The mid-level clouds of CR5 and the alto- and nimbo-type clouds of CR4 have a considerably smaller amplitude in their diurnal cycle since they are driven by solar heating to a much smaller extent.

The three cloud regimes consisting of high-level clouds (namely CR1, CR2 and CR3) present, for the biggest extent of the day, the lowest RFO values compared to the other cloud regimes and an overall small variability. CR3 is associated to high and very optically thick clouds, such as deep convective clouds and frontal convection. Its RFO values grow while the day progresses, forming a slight plateau of increased values around noon, which can arguably be connected to thermal convection leading to deeper clouds. In addition, CR2, which contains mostly cirrostratus and some thicker high clouds, gradually increases and peaks after the peak of CR3 is formed. Finally, the thin cirrus clouds of CR1 increase in frequency after noon and peak in the evening, reaching RFO values up to 8.5%, after the

peaks of CR2 and CR3. Provided that the deep convective cloud regime CR3 peaks earlier in the day than the cirrus and cirrostratus clouds of CR1 and CR2, respectively, this pattern could be indicative of deep convection and its related cirrus and cirrostratus remnants.

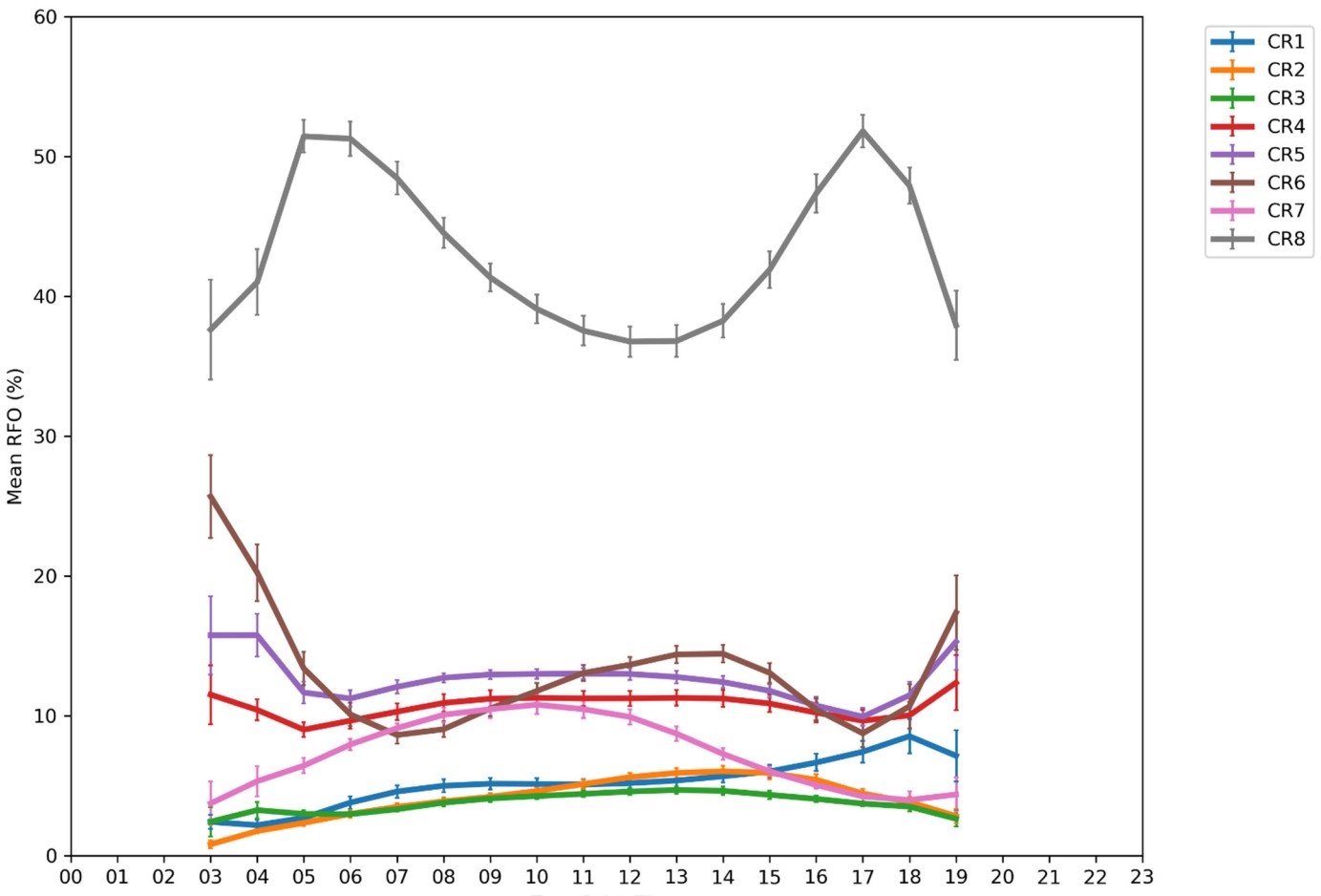

**Figure 5.** Mean diurnal variability of the relative frequency of occurrence (RFO) of the 8 CRAAS cloud regimes (CRs) over Europe for the 2004–2017 period.

Aside from the intra-annual and the diurnal variability of the cloud regimes, the 14 years of data from CRAAS allow for an in-depth study of changes in the frequency of occurrence of the derived cloud regimes both over the whole region of study and also locally, thus providing a useful insight into the changing climate of the region through the scope of the cloud regimes.

Moreover, the extensive time series of the cloud regime classification are used to supplement essential climate monitoring studies by identifying potential signals of specific substantial events that occurred over Europe during the 2004–2017 period. For instance, several of the most extreme drought events in Europe, which are thoroughly listed by Spinoni et al. in [44], can also be distinguished in the time series, presented in Figure 6, of the absolute monthly anomalies of the RFO of each cloud regime averaged over the whole region. For the most extensive and intensive drought events, a larger positive anomaly in the frequency of occurrence of the fair weather CR8, which is a regime of low cloud fraction, can be observed. During the corresponding months of the drought events, the opposite signal is apparent for the cloud regimes with high cloud fraction and especially for those that are associated to clouds that can potentially produce precipitation. The alto- and nimbo-type clouds of CR4, along with the middle-level clouds of CR5, present the strongest negative anomalies for all the drought events, while negative anomalies of

generally smaller magnitudes are noted for the cumulus and stratocumulus clouds of CR6 and CR7 during the majority of the drought periods. The high-level CR1, CR2 and CR3 have, overall, relatively minor absolute anomalies of varying signs.

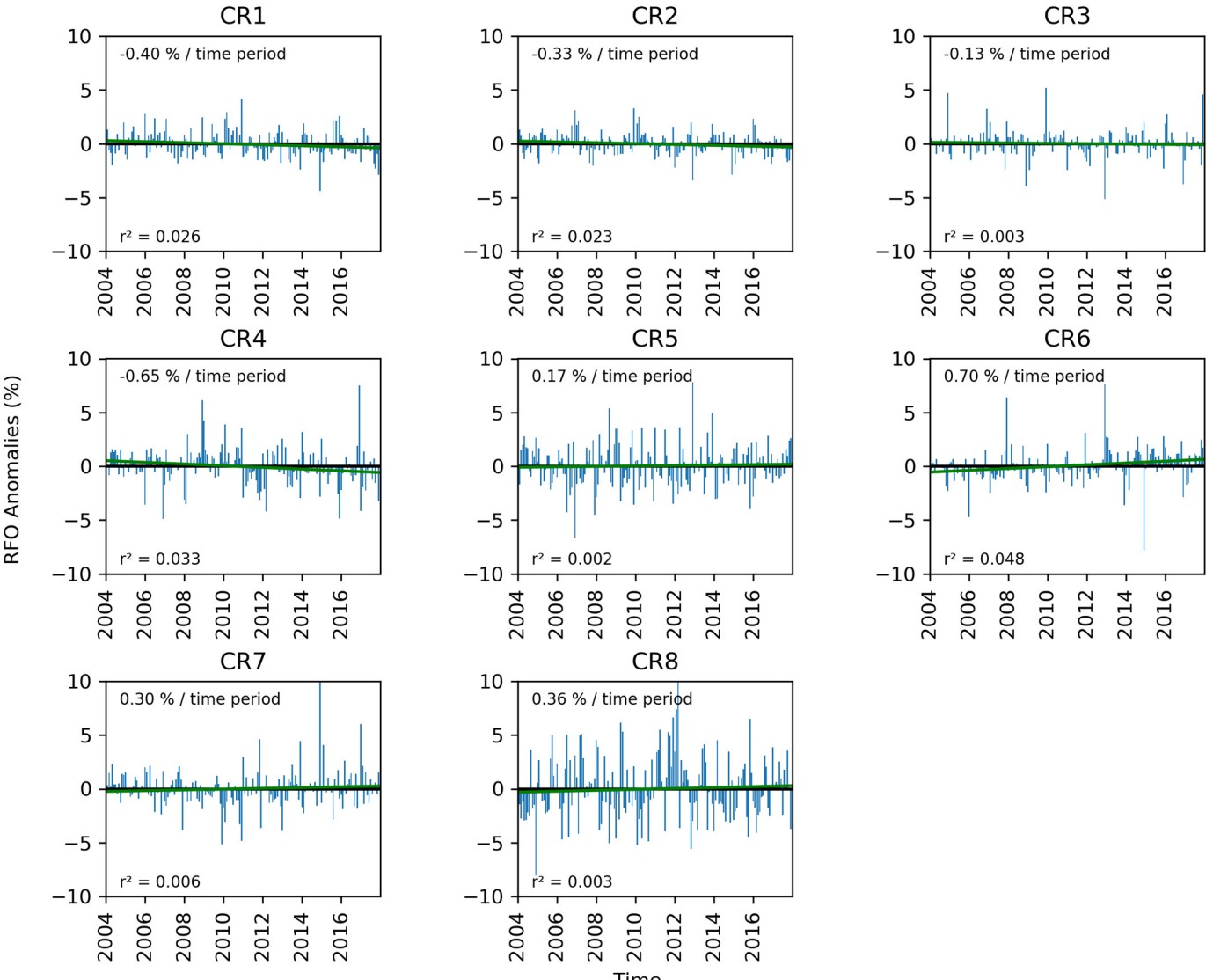

**Figure 6.** Inter-annual variability of the absolute monthly mean anomalies of the relative frequency of occurrence (RFO) of the 8 CRAAS cloud regimes (CRs) over Europe for the 2004–2017 period. The green line in each panel represents the linear regression fit for each time series.

To list a few of these events, according to Spinoni et al. in [44] there was a drought event peaking in July 2006 and covering approximately 34% of the region, including Southern Europe and areas around the Mediterranean, as well as regions of the Balkans and northeastern Europe. For the corresponding month, CR8 has a +5% anomaly, while the rest of the cloud regimes have negative anomalies, with CR5 having the largest one in magnitude ($-4.3\%$), except for cirrus and cirrostratus CR1 and CR2 that have small positive anomalies. In December of the same year, lasting until the spring of 2007, an event covering the Baltic Republics, Eastern Europe and later southeastern Europe can also be spotted in the time series. An additional large-scale drought reached its maximum in April 2011, covering 39.5% of the area including parts of South and Central Europe along with several Eastern European areas. For CR8 a +5.5% anomaly is observed for the corresponding month, while all the other cloud regimes, apart from CR3, express negative anomalies.

Bissolli et al., in [45], in a report from the German Weather Service, elaborate on the unusually dry weather throughout large parts of Europe in the spring of 2012. They discuss that the duration, the coverage and the intensity of the mentioned period was exceptional and that it was the continuation of a series of drought periods that took place during the previous year, 2011. This sequence of drought events is clearly distinguishable in the time series as a succession of large positive anomalies for CR8 around 2012, peaking at 10.4% in March of 2012. For the same month, CR4 exhibits the largest negative anomaly (−4.2%), followed by that of CR5 (−2.6%). For the remaining cloud regimes, negative anomalies of ranging magnitude can be detected as well, except for CR7, which has a minor positive anomaly of +1.3%. Naturally the signal of such events that are either on a smaller scale or of minor magnitude, as for example a drought in August 2005 affecting only 15.7% of the region referenced by Spinoni et al. in [44], is not always apparent in the time series of the whole European domain. However, further analysis focusing on more local aspects could provide additional insight and more detailed results on such topics.

Lastly, the trends, corresponding to the slope of the linear fit (green lines in Figure 6), of the eight cloud regimes during the 2004–2017 period are relatively small, with CR4 and CR6 having the most substantial change over the time series; CR4 exhibits the largest decrease (−0.65%) and, contrary to that, for CR6 the biggest increase (+0.70%) is observed. Trends were found to be statistically significant at a 95% confidence level for two of the high-level cloud regimes, CR1 and CR2 (−0.40% and −0.33% change in their RFO for the period of the study, respectively), for the alto and nimbo cloud regime CR4 and also for the low-level clouds of CR6.

The changes in the frequency of occurrence of the cloud regimes, shown in Figure 6 and discussed above, refer to averages over the whole domain of this study. Nevertheless, spatial patterns in the changing nature of the clouds and, by extension, of the cloud regimes over Europe, exist. The maps in Figure 7 represent the spatial distribution of the changes in the CRAAS-based cloud regimes during the 2004–2017 period, corresponding to the slope of the linear fit applied to the time series of the monthly anomalies for each $1° \times 1°$ grid cell. Dotted areas represent regions where the trend was found to be statistically significant with a 95% confidence level. Mixed patterns of relatively smaller changes in the frequency of occurrence of the three high-level cloud regimes (i.e., CR1, CR2 and CR3) can be observed over most of the domain. Only for the more convective clouds of CR3 can a slightly more extensive increase in their frequency of occurrence over higher latitudes, and especially over continental regions, be noted. However, over most of the region, these changes do not appear to be statistically significant. Areas of statistically significant trends are mostly found for CR1 and CR2 over the Central Mediterranean. CR4, containing mostly alto- and nimbo-type clouds, presents a widespread decrease over most of Europe with values reaching up to −2.8%. However, small regions of increasing frequency of occurrence for CR4, for instance, over the Bay of Biscay, exist as well. These trends are statistically significant over lower latitudes and parts of Eastern Europe. A considerably more diverse pattern of changes arises for the mid-level clouds of CR5. For the majority of the lower latitudes, and particularly over the marine areas of the Mediterranean, CR5 becomes less frequent, while for regions over Central and Eastern Europe CR5 becomes more frequent; locally CR5 increases with values up to +4.1%. The decrease in the frequency of occurrence of CR5 over most of the Mediterranean, as well as the increases over the less extensive areas of Central and Eastern Europe, were found to be statistically significant. A contrasting pattern to that of CR5 is discernible for the shallow cumulus of CR6. For most of the oceanic and coastal regions of Europe, CR6 shows large statistically significant increases in its frequency of occurrence, while, over land areas, decreased values can be noted. In particular, over the Central and Eastern Mediterranean, some of the biggest increases compared to all other cloud regimes are observed for CR6. Locally, these changes reach a maximum of +6.5%. A widely opposite pattern of, however, more minor changes is noticeable for the stratus and stratocumulus clouds of CR7. For CR7 no widespread statistically significant changes are apparent. The fair-weather cloud regime, CR8, has a diagonally contrasting pattern

of changes in its frequency of occurrence, with an extensive decrease over the northern parts of Europe and a widespread increase being apparent for the southern parts of the region. The largest decreases are located over Northern Germany and Denmark, with values reaching up to −4.7%, and over parts of North Africa and the Mediterranean, where an increase is observed, a maximum of +6.6% is approached. Nonetheless, exceptions in this general pattern exist, as, for instance, over the Eastern Mediterranean where the fair-weather cloud regime is regionally decreasing given the strong increase in the shallow cumulus of CR6. For CR8, statistically significant trends are apparent for a large area over the southern regions where extensive increases are observed and also over the decreases in RFO over Denmark and the North Sea.

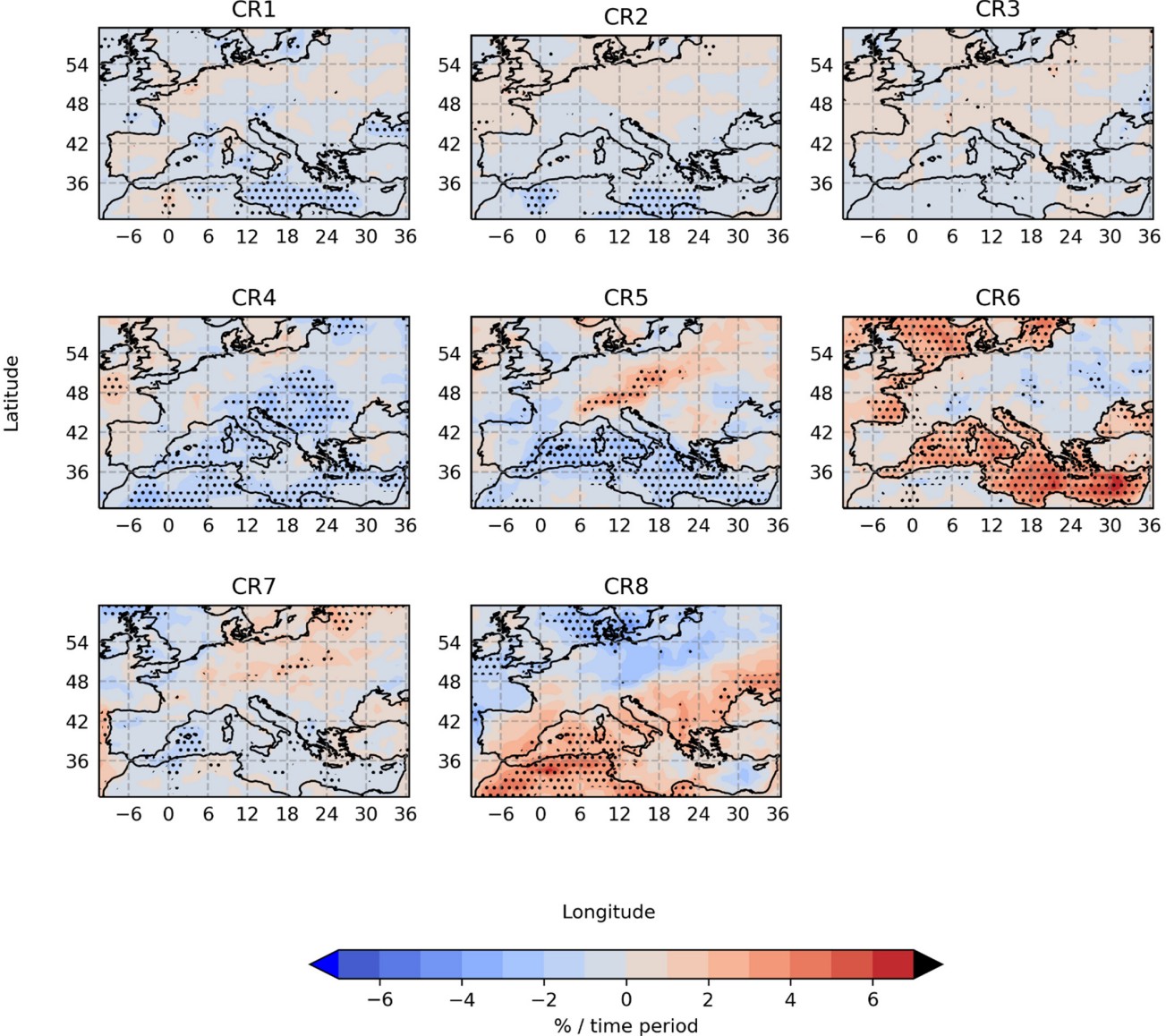

**Figure 7.** Changes in the relative frequency of occurrence (RFO) of the 8 CRAAS cloud regimes (CRs). The changes correspond to the slope of the linear regression fit of the absolute monthly mean absolute anomalies of the RFO for each $1° \times 1°$ grid cell during the 2004–2017 period. Dotted areas represent statistically significant trends, with 95% confidence level. The $r^2$ value of the linear regression fits averaged over the domain, for each cloud regime, ranges from 0.006 to 0.03.

In order to better understand the changes in the cloud regimes over the climate of Europe, Figure 8 shows the corresponding changes in the total cloud fraction. The principal

feature in the pattern of changes in the cloud fraction is an opposite pattern to that of the changes in the frequency of occurrence of the fair-weather regime (CR8). As expected, generally, over the areas where CR8 is increasing, the cloud fraction is decreasing given that less cloudy scenes dominate these regions. In contrast, locations over which CR8 is becoming less frequent, mostly present an increase in cloud fraction. An additional conclusion from the two figures is that, despite the large increase in the frequency of occurrence of CR6 over the Central and Eastern Mediterranean, the cloud fraction is mostly decreasing or, over less extensive areas, slightly increasing. This feature could be explained by the fact that CR6 contains mostly shallow cumulus and broken stratocumulus clouds, thus scenes of a widely reduced cloud fraction. This can also be noted in the total cloud fraction of CR6 (67.8% in Figure 1), which is the second lowest total cloud fraction of all the cloud regimes, after the fair-weather cloud regime, CR8. Therefore, while CR6 is becoming substantially more frequent over these regions, scenes of a low cloud fraction are becoming more frequent, leading to the decreasing cloud fraction apparent in Figure 8. In contrast, due to the considerably more minor increases in the more cloudy regimes (in terms of total cloud fraction) CR3 and CR4 (having total cloud fraction values of 99.7% and 93.5%, respectively), for example, over the Atlantic and the Bay of Biscay, a larger impact in the change in the cloud fraction can be observed. Finally, over Central Europe, the combination of the increasing middle-level CR5 and the decreasing fair-weather CR8, while the rest of the cloud regimes have slightly smaller changes, leads to a considerable increase in the cloud fraction over these areas.

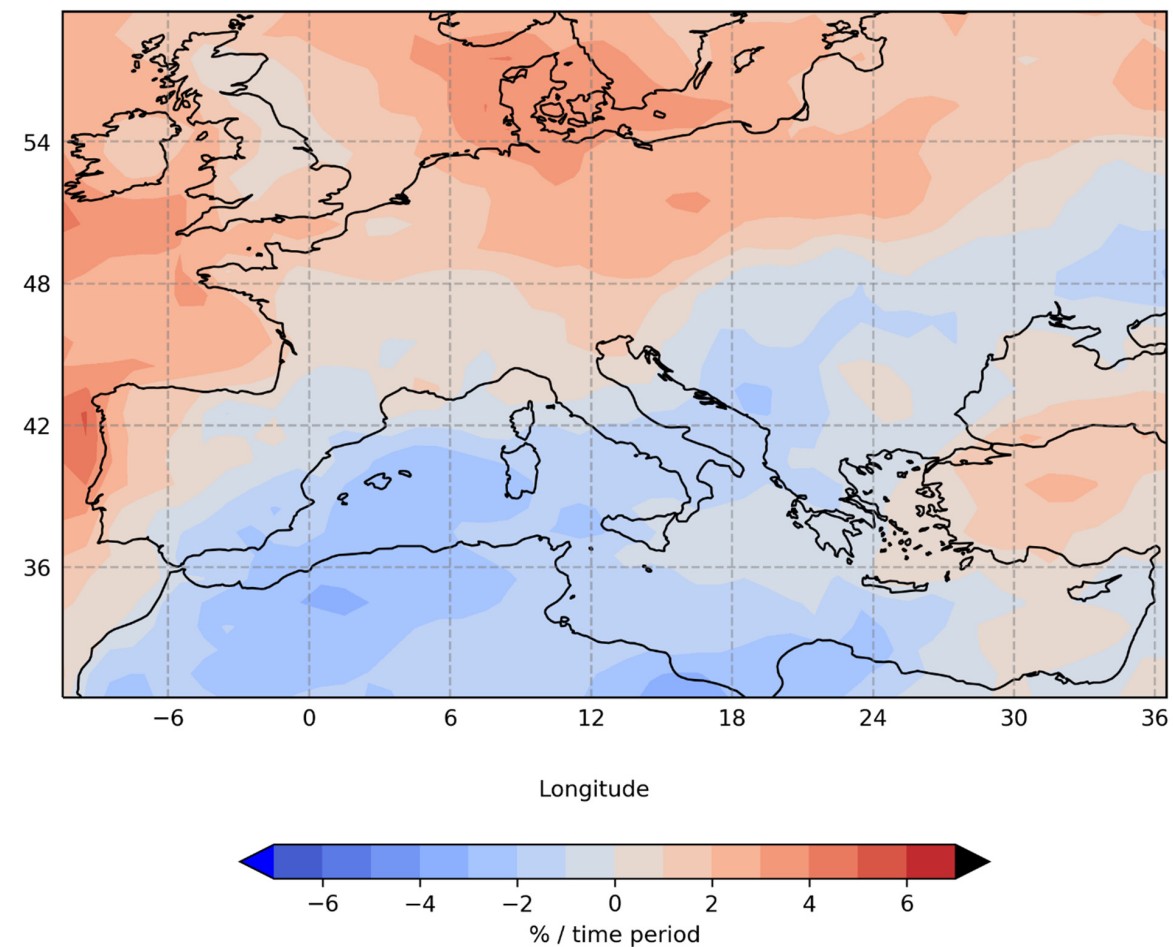

**Figure 8.** Similar to Figure 7 but for the total cloud fraction (CF) of the CRAAS dataset. The $r^2$ value of the linear regression fits averaged over the domain is equal to 0.011.

### 3.2. Connection of the Cloud Regimes to Large-Scale Weather Patterns

In this section an attempt to associate the occurrences of the cloud regimes to specific, predefined, large-scale weather patterns is provided. Occurrences of daily weather types derived from the Objective Weather Type Classification, produced by the German Weather Service, are compared to daily averaged cloud regime instances for the region of Germany (exact coordinates of the region are mentioned in the first section). By comparing large-scale weather patterns to cloud regimes, useful insight can be gained by understanding which cloud types can be mostly anticipated as a consequence of specific patterns of large-scale atmospheric circulation.

#### 3.2.1. Variability of the Cloud Regimes over Germany

Taking into consideration that the climatological variability of the cloud regimes is region-dependent [7], the overall mean annual and diurnal cycles over Germany of the CRAAS-based cloud regimes are shown in Figures 9 and 10, respectively, in order to gain a more in-depth comprehension of their regional behavior before associating them to specific weather types.

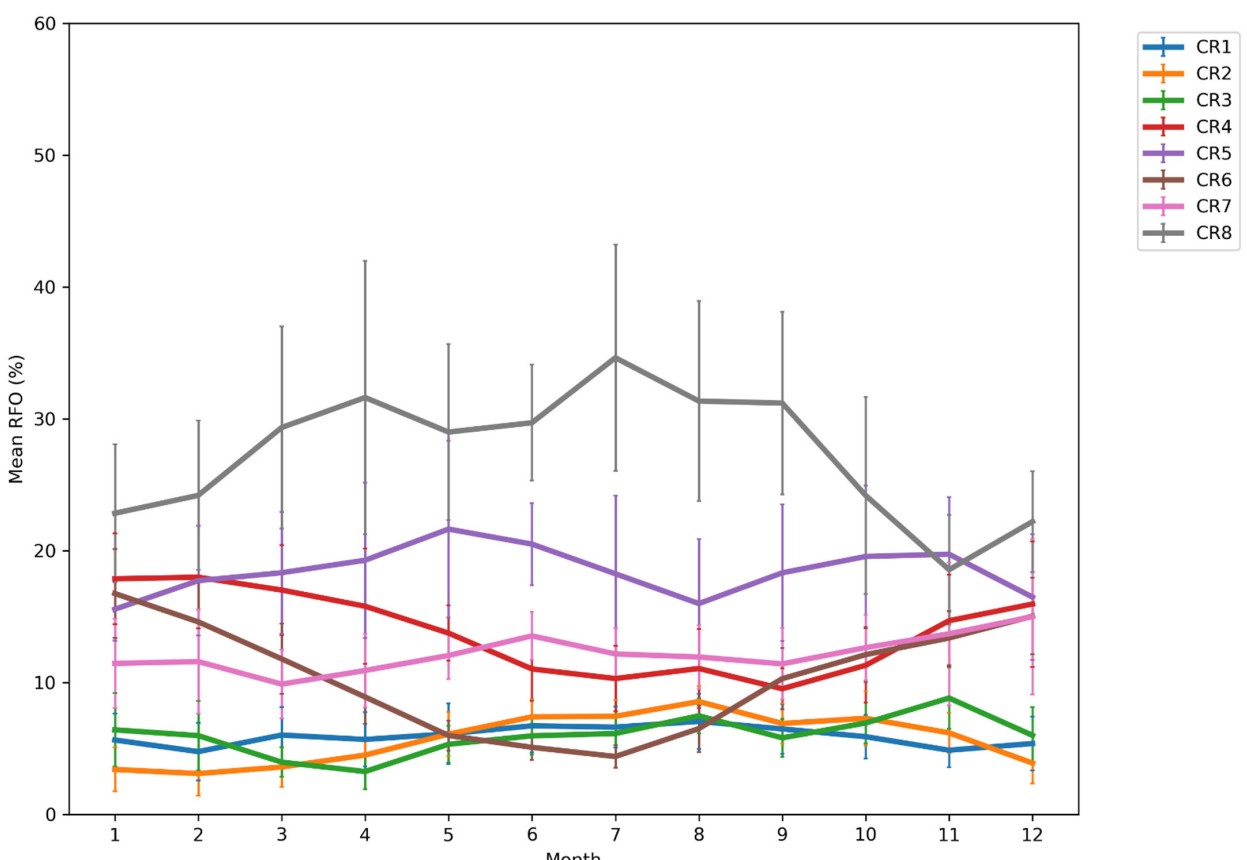

**Figure 9.** Similar to Figure 4 but for the region of Germany.

Compared to the intra-annual variability of the frequency of occurrence of the cloud regimes over Europe, the fair-weather CR8 has a significantly less intense variability, reaching a maximum of 34.6% in July, and a minimum of 18.6% in the much cloudier month of November. Moreover, CR8 has, overall, much smaller frequency of occurrence values over Germany compared to those for the full domain. The three high-level cloud regimes (namely CR1, CR2 and CR3) present, approximately, a similar variability, with a slight increase in their frequency of occurrence during summer. The cloud regime closely associated to frontal convection, CR3, peaks in November with 8.8%. The alto- and nimbo-type clouds of CR4 are considerably reduced from late spring until autumn, reaching

a minimum of 9.5% in September. In winter, however, CR4 nearly doubles its frequency of occurrence. CR5, which contains middle-level clouds, is the second most frequent cloud regime throughout the largest part of the year. Two peaks are clearly apparent for CR5, with RFO values of 21.6% and 19.7% in May and November, respectively. For the shallow cumulus CR6, a very intense annual cycle can be noticed, with an amplitude of 12.4%, due to large differences between summer and winter, which are the periods of its minimum and maximum values. CR6 is least frequent in July with 4.4%, while January is the month of its highest frequency of occurrence (16.8%). The second of the two low-level cloud regimes, CR7, which contains the thicker low-level clouds, has, generally, a weaker annual cycle, with values increasing mostly in autumn and early winter.

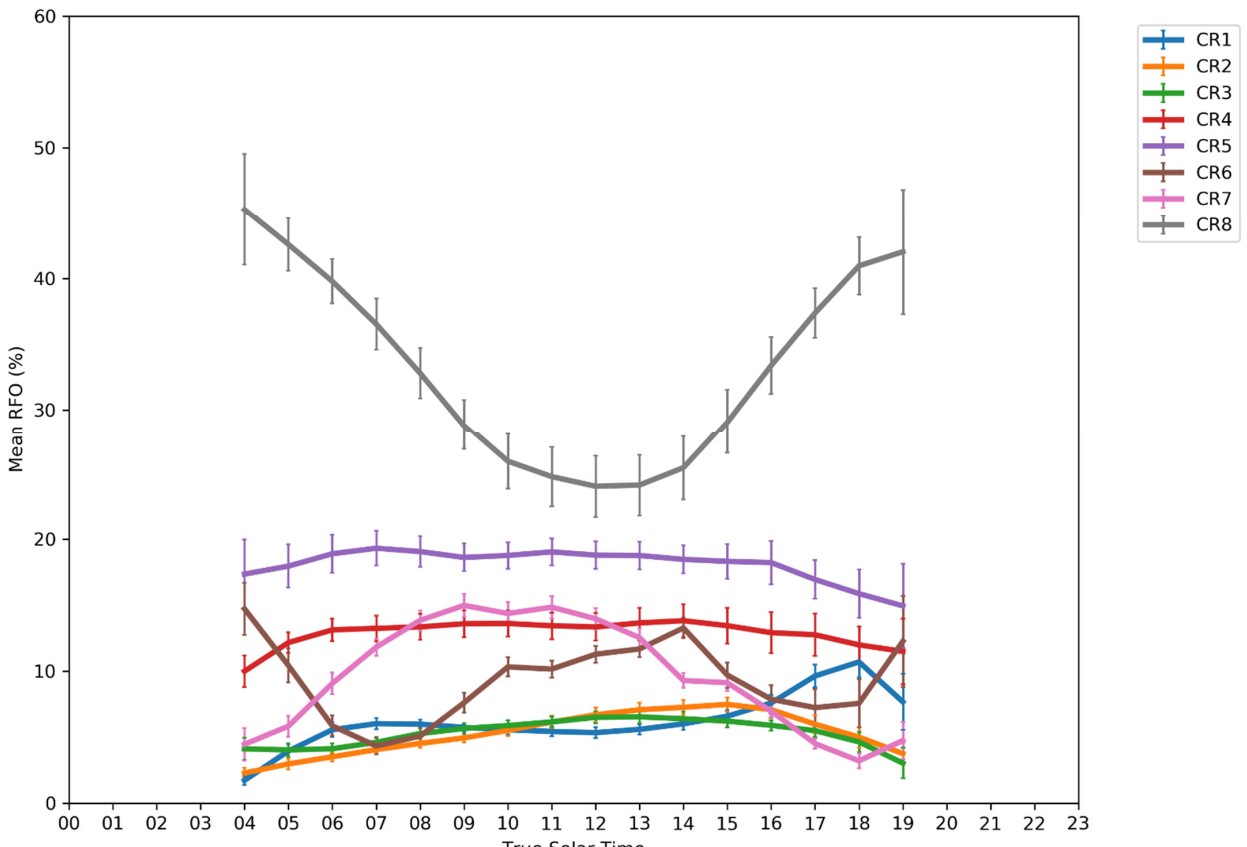

**Figure 10.** Similar to Figure 5 but for the region of Germany.

Regarding the diurnal cycle of the derived cloud regimes, as expected, the fair weather CR8 is more frequent in the early morning and in the evening, reducing its RFO values along the progress of the day, towards noon, while the cloudier regimes are becoming more frequent due to cloud formation. The three high-level cloud regimes, CR1, CR2 and CR3, are the least frequent during most of the day, only to increase their frequency of occurrence after noon. In particular, the thin cirrus, represented mainly by CR1, peak later in the day (reaching a maximum of 10.7%) than CR2 and CR3 that are associated to thicker high-level clouds, which again can be a potential indication of residual cirrus after deep convection. A slight differentiation to the corresponding diurnal cycle for the whole European domain is apparent for the alto- and nimbo-type clouds of CR4, as well as the middle-level clouds of CR5. The amplitude of their diurnal variability is considerably smaller, with their RFO values being seemingly constant for the largest part of the day, decreasing only slightly towards the evening when these cloud types start to dissipate. The early morning peak of the thin low-level clouds of CR6 can be attributed to radiation fog. With the progress of the day, while convection takes over, the formation of shallow cumulus clouds becomes more

frequent, peaking at 13.3%. CR7, consisting mostly of stratocumulus and stratus clouds, exhibits the expected diurnal cycle for these cloud types, being significantly more frequent in the morning (maximum RFO values up to 15%) and then sharply decreasing to 3.2% in the late evening.

### 3.2.2. Co-Occurrences of Cloud Regimes and Weather Types

The Objective Weather Type Classification is produced by the German Weather Service and provides a characterization of distinct weather situations based on several meteorological variables and criteria. The method uses results of the operational numerical weather analysis and forecast system of the DWD. It assigns, once daily, the weather situation over Central Europe to one of 40 possible weather types. The daily data basis is given by the numerical weather analysis of one of the operational forecast models of DWD at 12 UTC [25]. The meteorological input quantities used as the basis for the derivation of the weather types are the geopotential height, the temperature and the relative humidity of the 950, 850, 700, 500 and 300 hPa levels, as well as the horizontal wind components of the 700 hPa level [25]. The definition of a weather type for a certain day is based on the so-called OWTC indices, which are computed for various pressure levels from the model grid point data. These indices are the cyclonality index at the 950 and 500 hPa levels, the wind index at the 700 hPa level and the humidity index for the 950–300 hPa layer. The evaluation of these indices (e.g., distinguishing between dry and humid by setting a threshold of the humidity index) delivers a discrimination into classes and the combination of these classes (for wind, cyclonality and humidity) yields the corresponding weather type [25]. In total, 40 weather types are derived by combining the wind index, the cyclonality index and the humidity index. Each weather type is represented by a 5-letter name. The first two letters stand for the wind index (XX: no prevailing wind direction, NE: northeast, SE: southeast, SW: southwest, NW: northwest), the following two describe the cyclonality index (C: cyclonic, A: anticyclonic) for the near-surface and the middle of the troposphere, respectively, and the last letter defines the humidity index (W: wet, D: dry). A more detailed description of the Objective Weather Type Classification is available in [25].

Figure 11 shows the contingency table between occurrences of daily weather types of the Objective Weather Type Classification and daily averaged cloud regime instances over Germany for the common time period of 2004–2017. The side bar graphs represent the overall mean frequency of occurrence of the two classifications. Weather types driven by the prevailing westerlies of the middle latitude areas are apparently the most dominant for the region of focus, where the most frequent weather types are those consisting of either northwest or southwest wind indexes. As a result, there are weather types (for instance, those with a northeastern wind direction), with very low frequencies of occurrence, that are rarely encountered over the region. An example of weather types driven by the prevailing westerlies is the SWCCW weather type, with a southwestern wind direction, cyclonic index for both the near-surface level and the middle of the troposphere, and wet humidity index, which is a typical frontal weather pattern over Germany producing a considerable amount of deep convective clouds along with numerous altocumulus, altostratus, nimbostratus, as well as mid-level clouds. For this specific weather type, an increased amount of co-occurrences with CR3, CR4 and CR5, which are associated to frontal convection, alto- and nimbo-type clouds, and mid-level clouds, respectively, is observed. Slightly contrasting with the SWCCW weather type is SWCAW, with a similar wind direction, cyclonality index at the ground level and humidity index, but with anticyclonic conditions at the 500 hPa level. This weather type is substantially less convective and the prevailing anticyclonic pattern in the middle of the troposphere results in the prevention of the development of very optically-thick clouds. The SWCAW weather type therefore produces significantly more thin cirrus and generally more fair-weather clouds than SWCCW. This contrast is clearly apparent in the contingency table of Figure 11 with higher co-occurrences between SWCAW and CR1, CR2 and CR8 (being connected mostly to cirrus, cirrostratus and fair-weather optically thin clouds, respectively) than SWCCW and the same cloud regimes. A very common large-scale

weather pattern over Germany has northwestern flows, anticyclonic conditions at the near-surface level and increased cyclonality at the 500 hPa level accompanied by dry conditions within the 950–300 hPa layer. This specific weather type, namely NWACD, produces mainly subsidence clouds; thus it can be connected to a large amount of stratocumulus instances. Consequently, increased co-occurrences of NWACD and the low-level CR6 and CR7 cloud regimes can be distinguished in the contingency table of Figure 11. Stratocumulus clouds are largely included in CR7; however, a substantial fraction of cloud properties associated to stratocumulus clouds is included in CR5 as well (Figure 1). Hence the high co-occurrences between NWACD and CR5. Lastly, a typical anticyclonic, fair-weather weather pattern over central Europe and Germany is represented by the XXAAD weather type. XXAAD has no prevailing wind direction, for both atmospheric levels there is an anticyclonic index and reduced humidity conditions are predominant within the atmosphere. Therefore, the XXAAD weather type is a representative state of the atmospheric conditions over Germany connected to fair-weather clouds, thus resulting in numerous CR8 instances.

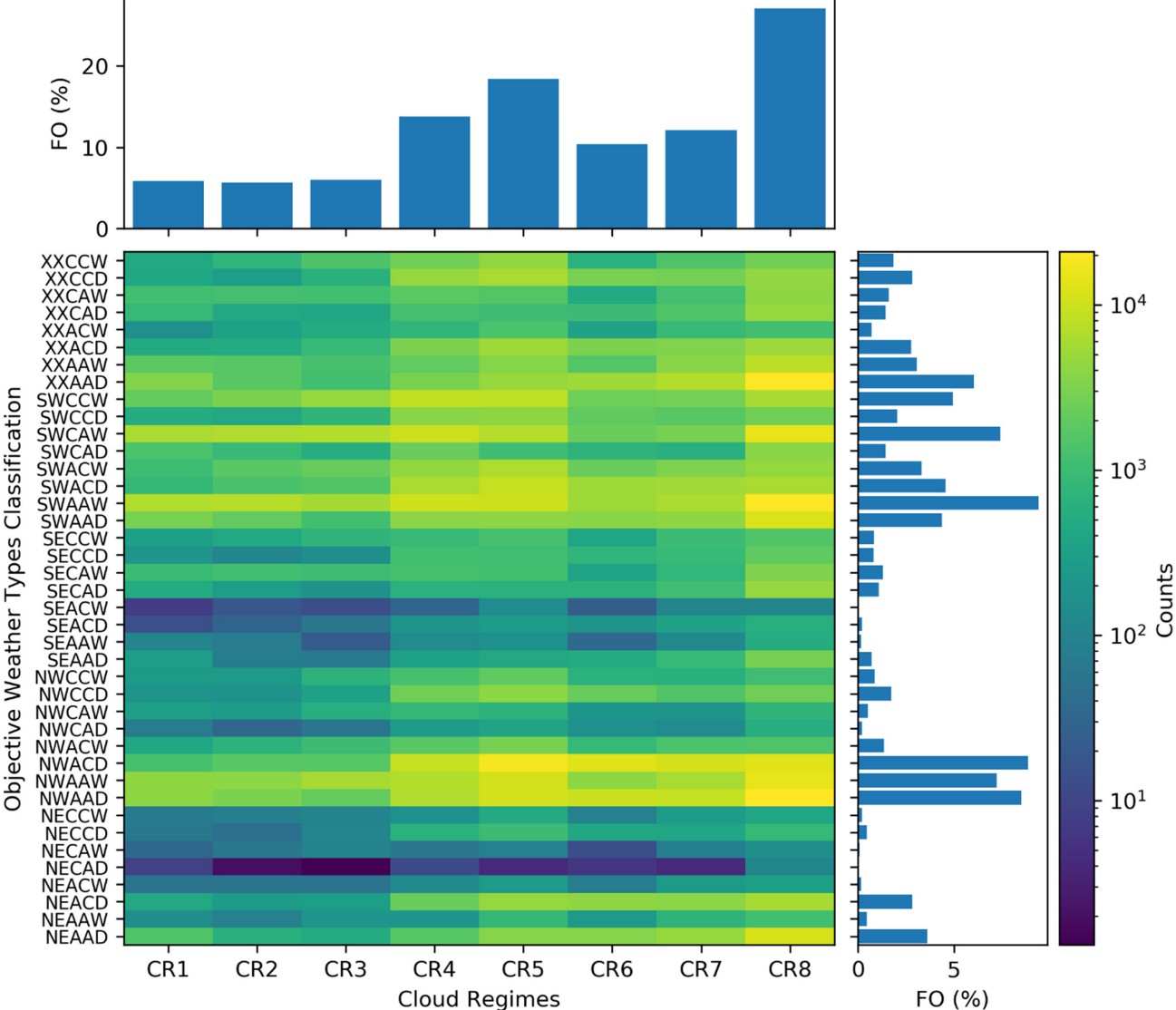

**Figure 11.** Contingency table between occurrences of the 40 weather types (WTs) of the Objective Weather Type Classification and the 8 CRAAS cloud regimes (CRs) over Germany for the 2004–2017 period. The two side plots show the relative frequency of occurrence (RFO) of the 40 WTs and the 8 CRs.

## 4. Summary and Conclusions

In the present study a well-established and robust method for cloud regime classification was applied to a long-term dataset of SEVIRI cloud properties, in order to derive cloud regimes over Europe and to study their climatological variability as well as changes in the cloud regimes during the period of the study. In addition, relations between particular large scale weather patterns and the formation of specific cloud types, and, by extension, the derived cloud regimes, were examined. In particular, the k-means clustering algorithm was applied to joint cloud histograms of cloud top pressure and cloud optical thickness, generated from the corresponding products of the CLAAS-2.1 climate data record over the region of Europe and covering the 2004–2017 period. The constructed joint cloud histograms, along with the centroids of the eight cloud regimes and the classification labels for each $1° \times 1°$ grid cell and each 15-min time step, are available in the Cloud Regime dAtAset based on the CLAAS-2.1 climate data record (CRAAS) [37].

Eight cloud regimes were derived in order to describe the nature of the clouds over Europe. More specifically, three of the generated cloud regimes were associated with high-level clouds of increasing optical thickness (from thin cirrus to deep frontal convective clouds), one regime was mainly connected to alto- and nimbo-type clouds, a fifth cloud regime was found to mostly represent middle-level clouds with primarily continental characteristics, two low-level cloud regimes depicted thinner broken clouds and thicker low-level clouds, respectively, and one cloud regime consisted mostly of fair-weather clouds and of joint cloud histograms that did not have any specific shape to form a particular category. Taking advantage of the available long time series and the detailed sampling cycle of SEVIRI (15-min), the annual and diurnal cycles of the CRAAS cloud regimes were calculated in order to provide a detailed representation of the climatology of clouds over Europe through the scope of the cloud regimes. In addition, the effectiveness of such a classification for climate monitoring was proven by the ability to identify specific extreme events in the time series of the anomalies of the cloud regimes. Changes in the frequency of occurrence of the cloud regimes were investigated in order to provide additional insight into the changing climate of the region. The largest changes were found for CR4, the cloud regime connected to alto- and nimbo-type clouds, for which a widespread decrease in its frequency of occurrence was observed over most of Europe. Additionally, for CR6 large increases over the marine areas were found. Statistically significant trends, at a 95% confidence level, were computed for CR1, CR2, CR4 and CR6. Taking into account the overall characteristics of the cloud regimes, the features of the changes in the frequencies of occurrence were complemented by changes in the cloud fraction to provide a complete picture of the changing clouds over the region of the study. Focusing further on more regional variability of the cloud regimes, the variability of the cloud regimes over the region of Germany was studied and, also, over the same region, connections between large-scale weather patterns and the cloud regimes were investigated. For this relation, co-occurrences of 40 distinct weather types from the Objective Weather Type Classification and the cloud regimes were studied. The biggest co-occurrences were calculated for weather types with a western wind direction. For instance, SWCCW appears frequently with the appearance of clouds associated to CR3, CR4 and CR5. Another weather type driven by the prevailing westerlies, NWACD, was found to be accompanied by increased appearances of the low-level cloud regimes, CR6 and CR7.

The concept of the cloud regimes has been proven useful and able to enlighten different aspects of numerous types of studies. They can be applied to calculate transition probabilities between the different cloud regimes, and they can also assist in the cloud representation and diagnostics of weather and climate models. Radiative effects associated to specific cloud regimes can be computed as well (as shown for example in [6]), providing an interesting approach to radiation studies, e.g., computing long-term cloud radiative effects from historical cloud regime observations. In addition, constant climate monitoring on a regional or more extended scale can be facilitated by observing the changing environment through the cloud regimes. Improved continuations of the cloud regime datasets

can help in the extension of the time series, resulting in even more long-term climatologies of such classifications. For instance, the improved and extended follow-up version of the CLAAS-2.1 climate data record (CLAAS-3) could assist with refinements of the required cloud optical properties and also with a continued time series up to the present day. Moreover, additional analysis studying the trends over specific climate zones or locations could assist in exploring more detailed patterns of changes. The effect of using different grid sizes in cloud regime studies could also provide additional insight into understanding the cloud regime concept. Finally, an inter-comparison study between different cloud regime datasets could contribute to improved knowledge of the cloud regime concept by better understanding the existing differences between the distinct cloud regimes sets.

**Author Contributions:** Conceptualization, V.T., A.H. and A.M.; methodology, V.T. and A.H.; software, V.T. and A.H.; validation, V.T., A.H. and J.T.; formal analysis, V.T.; investigation, V.T.; resources, A.H. and M.S.; data curation, V.T., A.H. and M.S.; writing—original draft preparation, V.T.; writing—review and editing, V.T., A.H., M.S., J.F.M., N.B., J.T. and A.M.; visualization, V.T., A.H. and J.F.M.; supervision, A.H. and A.M.; project administration, A.H. and A.M.; funding acquisition, A.H. and A.M. All authors have read and agreed to the published version of the manuscript.

**Funding:** This research was supported by the Leibniz Institute for Tropospheric Research, grant number 4-7323/29/5-2021/2017 from 20 January 2021. The APC was funded by the Leibniz Institute for Tropospheric Research.

**Data Availability Statement:** The data presented in this study are openly available in Zenodo at https://doi.org/10.5281/zenodo.7120267. (accessed on 20 September 2022).

**Acknowledgments:** The authors would like to thank Hartwig Deneke for his contribution in the discussions and technical support in several stages of the present work.

**Conflicts of Interest:** The authors declare no conflict of interest. The funders had no role in the design of the study; in the collection, analyses, or interpretation of data; in the writing of the manuscript; or in the decision to publish the results.

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
