# Peer review of "CRAAS: A European Cloud Regime dAtAset Based on the CLAAS-2.1 Climate Data Record"

_remotesensing, doi:10.3390/rs14215548_

Round 1
Reviewer 1 Report
The paper presents an extensive analysis of satellite-derived cloud imagery. It is an interesting analysis of an overwhelmingly large set of measurements, and the authors have distilled many interesting findings. The paper should be published with a few questions spelled below addressed:
1) No mention of uncertainty:
Fig 3b shows a 1x1 deg CLEAR (cat 8) categorization in the middle of a frontal cloud band that stretches from southwest - northeast France. Clearly this is an error, as the satellite imagery (Fig 3a) shows no clear area.
There must be some mention of the magnitude of uncertainty, especially for many of the ambiguously-distinguished categories.How many of these 1x1 degree assignments are wrong or ambiguous? is it 1/1000 or 1/10?
2) Figs 6-7 shows linear trends of the cloud categories 2004-2018. Clearly the trends are really small compared to the month-to-month and annual variability. The authors MUST present the R^2 correlation coefficients for the linear fits on fig 6. For Fig 7, maybe the average R^2 over the domain should be presented on each figure? Mmaybe the caption should state something like "R^2 values for the regressions ranged from 0.001 - .01" (substitute actual values) where something like the 10th - 90th percentiles of the R^2?
3) for Fig 8 I have the same request as above. Here you could actually plot a contour diagram of the R^2 values. at a minimum there Must be a mention of the R^2 averaged over the domain in the caption.
4) There must be some mention about the horizontal spatial resolution dependence of this analysis. What is the resolution of the raw pixels? How many cloud top temperatures go into each 1x1 degree grid? If this analysis were repeated at 2 or 5 degrees grids, would results change? What about going to a 0.5x0.5 degree grid?
Reviewer 2 Report
The manuscript describes the generation of a database of cloud regime (CR) classes using the cloud feature climatology from Seviri on MSG and applying a k-mean cluster analysis to the histograms of cloud optical thickness vs. cloud top pressure. After the method description the authors interpret and discuss the classification result. The established classes were then explored wrt. their spatial and temporal distribution. The 14year observation period allowed for the discussion of climatological trends in the data. Finally the authors explore the statisics for the subregion of Gemany a relate their CR classes to a weather pattern classification.
The paper is a convincing contribution to the CR classification and application. It completes existing works for other parts of the world and other remote sensing instruments. The presented statistic results demonstrate the high potential of the CR database.
The paper is well written, the material is presented adequately, the discussion and conclusions are sound. The graphical material is thoroughly designed and prepared with high quality.
I recommend the paper for publication in the 'Remote Sensing' after minor revision of the issues given below.
Major points:
1* Some additional information on the method design would be useful for the reader. E.g. Give the number of Seviri pixels in the grid cells.
How where the bin widths for COT and CTP selected? The CLASS uses finer bin widths.
The authors should also describe and discuss the differences and possibly the novelties of their method to the methods applied in the existing papers on CR classification.
2* Comments to the classification:
CR2-CR3 are well separated in the JCH by their different COT, but show very similar behavior in all the statistics.
I wonder, if the naming of CR3 as 'deep convective clouds' is appropriate. The clouds are dense but not 'deep' in their vertical extension.I suppose this regime contains dense cirrus anvils from deep convection but also cirrus associated with frontal systems (as the authors describe e.g. in 413f)
CR6 might have mixed 2 cloud regimes fog and low clouds from shallow convection. The diurnal cycle shows the 2 separated processes of fog generation/dissolution and the afternoon shallow clouds. Can this possible merging of the 2 cloud regimes be a consequence of the coarser bins for CTP in CRAAS vs., CLAAS?
Minor issues:
# It might be welcomed by readers less familiar with the European geography to see a picture showing the region of Europe with the subregion of Germany marked in the introduction. Also the size (in km) and/or the area of both regions could be given explicitly.
# 220-244: Can you explain, give a definition of the 'pattern correlation coefficient'
# 249ff A euclidean distance is used. How was it defined? Did you normalize the differences? The histogram edges are logarithmic spaced, and moreover the COT and CTP have strongly differing ranges.
# Fig.3 The authors might consider to give the same projection in both figure parts for better comparison, i.e. present the 3a) also in the cylindrical longitude-latitude projection.
# Diurnal cycle (Figs.5,10) Why is the considered time asymmetric with respect to noon? (3-19, 4-19 solar time)
# sec. 3.2.2 Line 707f: What is the 'daily averaged' CR for the Germany region? How is it defined?
# The manuscript contains several extended passages of plain text which make it hard to read. E.g. lines 302-361, 436-375, 481-534 and others. I suggest to present the content visually more structured.
# I also wonder if a schema or a figure could illustrate the method description in sec. 2.2.
Reviewer 3 Report
Please see the attachment.
